# Why do G-quadruplexes dimerize through the 5'-ends? Driving forces for G4 DNA dimerization examined in atomic detail

**Mateusz Kogut, Cyprian Kleist, Jacek Czub** *

Department of Physical Chemistry, Gdansk University of Technology, Gdansk, Poland

* jacek.czub@pg.edu.pl

**Data Availability Statement:** All input files for REUS simulations (initial configurations, topologies and input files for Plumed) and trajectories corresponding to free energy minima, saved every 20 ps and stored in the GROMACS xtc format are

## Abstract

G-quadruplexes (G4) are secondary structures formed by guanine-rich nucleic acid sequences and shown to exist in living cells where they participate in regulation of gene expression and chromosome maintenance. G-quadruplexes with solvent-exposed guanine tetrads show the tendency to associate together through cofacial stacking, which may be important for packaging of G4-forming sequences and allows for the design of higher-order G4 DNA structures. To understand the molecular driving forces for G4 association, here, we study the binding interaction between two parallel-stranded G-quadruplexes using all-atom molecular dynamics simulations. The predicted dimerization free energies show that direct binding through the 5'-G-tetrads is the most preferred of all possible end-to-end stacking orientations, consistently with all available experimental data. Decomposition of dimerization enthalpies in combination with simulations at varying ionic strength further indicate that the observed orientational preferences arise from a fine balance between the electrostatic repulsion of the sugar-phosphate backbones and favorable counterion binding at the dimeric interface. We also demonstrate how these molecular-scale findings can be used to devise means of controlling G4 dimerization equilibrium, e.g., by altering salt concentration and using G4-targeted ligands.

## Author summary

Native DNA usually folds to form the canonical double helix, however, under certain conditions, it can also fold into other secondary structures. Some of the most interesting ones are G-quadruplexes (G4)—compact DNA structures in which guanines assemble into multilayered tetrads, and whose formation has been reported at the ends of linear chromosomes (telomeres) and at different regulatory regions of the genome. Although structural and basic energetic properties, as well as some biological functions of G-quadruplexes are quite well understood, not much is known about their propensity to form aggregated structures. A very high density of G-quadruplexes at telomeres along with their large exposed planar surfaces indeed favor G4 aggregation through end-to-end stacking, which might be important for the protection of telomeres and DNA packaging. In this research, using computer simulations, we provide insight into molecular origins of

available from the figshare service, accessible at doi.org/10.6084/m9.figshare.8276867.v1 All MD parameters and other data required to reproduce our results are included in the Methods section and Supporting Information.

**Funding:** This work was supported by the Foundation for Polish Science (FNP) Homing Plus Programme, co-financed from the European Union's Regional Development Fund within the Operational Programme Innovative Economy [HOMING PLUS/2011-4/3]; Funding for open access charge: Gdansk University of Technology; Computational time: PL-Grid Infrastructure; Academic Computer Centre TASK. The funders had no role in study design, data collection and analysis, decision to publish, or preparation of the manuscript.

**Competing interests:** The authors have declared that no competing interests exist.

stability of the higher-order G-quadruplexes and explain in structural and energetic terms a strong preference for one particular end-to-end stacking orientation. Based on the recognized aggregation driving forces, we also suggest methods for controling the aggregation preferences openining up new opportunities for designing oligomeric G-quadruplexes.

## Introduction

G-quadruplexes (G4) are non-canonical four-stranded structures formed by guanine-rich sequences of nucleic acids, in which sets of four guanine residues associate into planar arrays (G-tetrads) through cyclic Hoogsteen hydrogen bonding [1–3]. By stacking on top of each other, G-tetrads make up the core of the G4 structure (see Fig 1A), and, in unimolecular G-quadruplex, are connected by three intervening loops of variable length and sequence. G-quadruplexes can fold into a variety of topologies, differing in the relative orientation of the four guanine runs and in the arrangement of loop regions [4, 5]. A delicate conformational equilibrium between these topologies depends on the length of the G-runs, the length and sequence of the loops, and the type and concentration of alkali metal cations which are known to be crucial for the G-quadruplex formation [6–9].

Recently, using structure-specific antibodies and in-cell NMR, DNA G-quadruplexes have been shown to occur throughout the genome in living cells [10–12], with a markedly higher density observed at the telomeric regions (up to 25% of all G4 DNA) [13]. Indeed, consisting of tandem repeats of the TTAGGG sequence and single-stranded 3'-overhang, vertebrate telomeric DNA is more likely to fold into G4 structures. G-quadruplex-forming sequences are also over-represented in other regulatory regions of the genome, including promoters [14], immunoglobulin switches [15], introns [16], and 5' untranslated regions [17]. Accordingly, G-quadruplexes are thought to be involved in regulation of DNA replication and transcription, genetic recombination, maintaining chromosome stability, and other fundamental cellular processes. With their unique molecular structure and biological significance, G-quadruplexes attracted much attention as drug-design targets [18]. In particular, since G4 structures have been shown to inhibit, among others, telomerase [19] and HIV integrase [20], there is reason to believe that selective G4-stabilizing ligands could act as anti-cancer or anti-viral agents [21, 22].

It has long been thought that, due to their layered structure, G-quadruplexes should show tendency to associate through stacking interactions between the external G-tetrads. This should be possible especially for parallel-stranded G-quadruplexes which have their external G-tetrads almost fully exposed to the solvent. As expected, the formation of higher order G4 structures was observed in a thermal denaturation study of parallel G-quadruplexes formed in a 'bead-on-a-string' fashion by long GGGT [23] and TTAGGG [24] DNA sequences. Furthermore, by using a combination of circular dichroism and mass spectrometry, it has been shown that aggregation of parallel G-quadruplexes (composed of GGGT repeats) depends upon the presence of flanking bases and therefore is likely to occur through stacking of the external G-tetrads, with G4 dimers and trimers being the most probable aggregated forms [25].

Analysis of dimerization equilibrium of the propeller-type G-quadruplexes with NMR spectroscopy confirmed the above finding and showed that out of the three possible end-to-end orientations (see Fig 1A), the 5'-5' stacking is strongly preferred [26]. Consistently, the 5'-5' stacking arrangement is also found in all high-resolution NMR structures of G4 DNA dimers

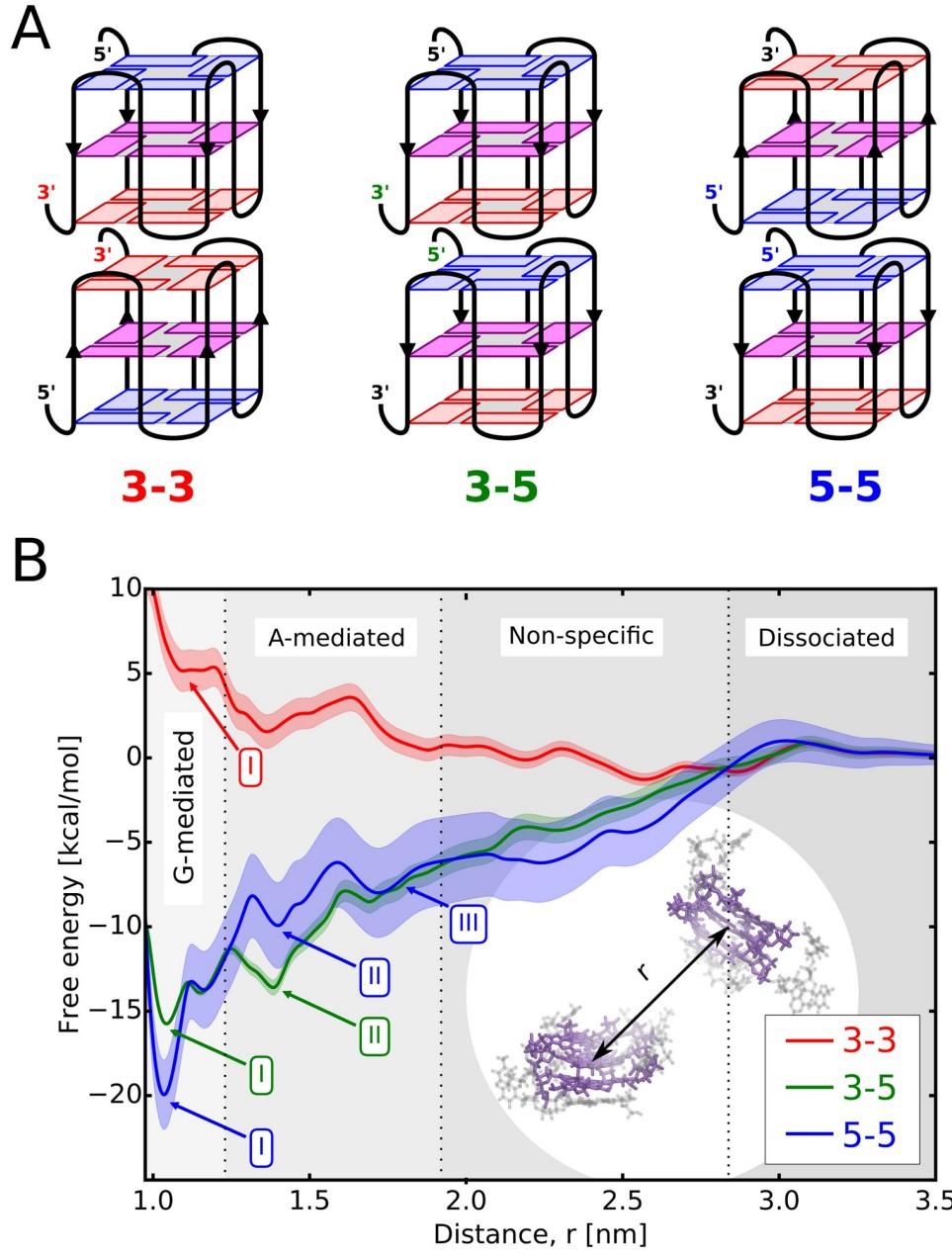

**Fig 1. G4 dimerization preferences.** A: Three possible end-to-end stacking orientations of the G4 units in the dimeric structure formed by two parallel G-quadruplexes: 3'-3', 3'-5' and 5'-5'. The 5'-terminal, middle and 3'-terminal G-tetrads are shown in blue, magenta and red, respectively. B: Free energy profiles for the formation of the above dimers as a function of the separation distance between the G4 units, $r$. The profiles were computed using CHARMM36; for the parmbsc1 results see S3 Fig. Shades regions represent consecutive stages of the dimerization process. The same color coding for the G-tetrads and dimerization modes is used in all other figures.

determined to date, including $(GGA)_4$ and $GIGT(GGGT)_3$ model sequences [27, 28], as well as the G4-forming regions of the *N. gonorrhoeae* pilE gene and the human CEB1 minisattelite [29, 30]. The preference for the 5'-5' mode is also reflected in the x-ray structures of parallel G-quadruplexes in which individual G4 units tend to stack with each other in the crystal lattice through the 5'-terminal G-tetrads [31–33].

It has been suggested that dimerization of G-quadruplexes provides additional stabilization of the individual G4 DNA units, and therefore might be important for packaging of G4-forming sequences [34, 35] preferentially adopting parallel conformations. However, predicting whether particular G-quadruplexes are prone to aggregation (e.g., dimerization) is far from trivial, because this process depends on several factors including the overall folding topology, the loop and flanking sequences and the concentration and type of cations [24, 26]. In addition, the dimerization equilibrium can be modulated by small-molecule ligands, such as meso-porphyrin and naphthalene diimide derivatives. By associating with the 3'-terminal G-tetrad, these ligands favor the formation of the parallel G-quadruplexes, leaving the 5'-terminal G-tetrad free for favorable interaction with the other G4 unit [32, 36]. Furthermore, some molecules, such as cationic porphyrins, trisubstituted acridines and berberines, were shown to increase the stability of G-quadruplex dimers by intercalating between interfacial G-tetrads [37–42].

Previous computational studies have shown that the preformed dimeric and trimeric assemblies of the parallel-stranded G-quadruplexes with 3'-5' stacking orientations are stable on the time scale of 15 ns [43]. MM-PBSA calculations further suggested a significant stability of the parallel G4 dimers relative to the monomeric state [44]. Absolute stacking energies for simple models of isolated G4 cores and stacking interfaces have also been determined by force-field and quantum chemical calculations combined with structural database search [45]. However, despite this research, the origin of the driving forces for the dimer formation and specifically of the strong preference for the 5'-5' stacking mode remains largely unexplored, making it difficult to predict and control the dimerization process [46]. Therefore, in the present work, we probe the determinants of the stability of G4 dimers by computing the free energy profiles for all possible $\pi$-stacked assemblies formed by the parallel telomeric G-quadruplexes, using explicit-solvent all-atom molecular dynamics simulations with a total simulation time of $\sim 100$ $\mu$s. These calculations correctly reproduce the predominance of the direct 5'-5' stacking mode over the remaining dimeric forms and show that dimerization equilibrium is governed largely by a competition between the electrostatic repulsion of the sugar-phosphate backbones and favorable counterion binding at the interface between the G4 monomers. We further show that this detailed knowledge can be used to predict shifts in the dimerization equilibrium in response to changes in salt concentration and addition of G4-binding ligands.

## Results

### Free energy simulations confirm the previously reported strong preference for the 5'-5' dimerization mode

To examine the relative stability of possible dimeric structures formed by the parallel telomeric G-quadruplexes, we computed the free energy profiles for the main dimerization modes defined by three distinct end-to-end stacking orientations: 3'-3', 3'-5' and 5'-5', hereinafter referred to as the 3-3, 3-5 and 5-5 modes (Fig 1A). As a collective coordinate describing the dimerization process, we used the separation distance between the centers of mass of the guanine cores of the two associating G-quadruplexes (inset of Fig 1B). The distributions of the relative orientations between G4 units (S1 Fig) confirm association through cofacial stacking but also show that at longer distances G-quadruplexes sample the entire range of relative orientations, possibly facilitating the free energy convergence (S2 Fig).

The free energy profiles, $G(r)$, in Fig 1B reveal unexpectedly large differences in the stability of the three dimeric structures, with the 5-5 mode being by ca. 4 and 24 kcal/mol more stable than the 3-5 and 3-3 modes, respectively. By assuming that translation and rotational entropy loss is similar for the three modes, we estimated the corresponding dimerization free energies

**Table 1. Estimated dimerization free energies.** $\Delta G^{\circ}_{dim}$ were estimated for the three considered dimerization modes from the free energy profiles in Fig 1B, separately for the G-mediated and A-mediated states, and are shown along with the corresponding equilibrium populations.

| Dimerization mode | $\Delta G^{\circ}_{dim}$ [kcal/mol] | | Eq. population [%] | |
|---|---|---|---|---|
| | G-med. | A-med. | G-med. | A-med. |
| 3-3 | 6.4 ± 1.1 | – | ∼0 | – |
| 3-5 | −13.5 ± 0.1 | −11.7 ± 0.3 | 0.1 | ∼0 |
| 5-5 | −17.5 ± 0.1 | −8.6 ± 1.8 | 99.9 | ∼0 |

(Table 1; see Methods) and found that the 5-5 orientation is assumed with 99.9% and 3-5 with only 0.1% probability, while the 3-3 mode is virtually absent from the dimeric ensemble. Importantly, a clear preference for the 5-5 mode seen in our simulations, both CHARMM36 and parmbsc1 (see S3 Fig), is fully consistent with the available experimental data. Indeed, by using NMR spectroscopy, it has been found that the parallel DNA G-quadruplexes composed of the GGGC, GGGT and GGA motifs (PDB codes: 2LE6 and 1MYQ) [26–28], as well as the human CEB1 minisatellite (2MB4) and the bacterial pilE (2LXV) G-quadruplexes [29, 30] show a strong tendency to form dimers via the 5'-5' stacking interface. Similar preference was observed for the human telomeric G-quadruplexes under molecular crowding conditions [47] and for TERRA G-quadruplexes [48, 49]. Interestingly, the propeller-type telomeric G-quadruplexes also crystallize preferentially in the 5'-5' stacking arrangement, as found in crystallographic studies [31].

The shape of the computed profiles with several free energy minima suggests diversity of dimeric structures, even for a given end-to-end orientation. Since the average distance between the adjacent G-tetrads in a guanine core is about 0.35 nm, the deepest minimum at 1.1 nm has to correspond to direct stacking between the terminal G-tetrads (states labeled as 'G-mediated' in Fig 1B). In turn, the position of the local minima at 1.4 and 1.7 nm suggest that adenine residues may also be involved in mediating dimer formation ('A-mediated' states in Fig 1B), especially as in telomeric G-quadruplexes, adenines tend to stack onto the 5'-G-tetrad [50, 51], and hence two minima are seen for the 5-5 mode (minima II and III) and one for the 3-5 mode (II) (see also detailed structural analysis below). For separation distances from 1.9 to 2.9 nm, we observed various poorly-defined and non-specific interactions between the two monomers, and beyond 2.9 nm G-quadruplexes are no longer in contact yielding a plateau in the free energy profile.

## Parallel G-quadruplexes dimerize almost exclusively via direct contact of guanine planes

To confirm the above characterization of the free energy minima and investigate how structurally diverse are different dimerization modes, we performed cluster analysis of the MD-generated unbiased dimeric ensemble, separately for each of the six minima indicated in Fig 1B. This was done using the hybrid k-centers k-medoids clustering algorithm for all heavy atoms of the two guanine cores, with a RMSD cut-off of 0.3 nm [52].

Fig 2A shows schematic representations of dimeric structures identified by cluster analysis in the individual minima along with their equilibrium contributions to a given state. As expected, the lowest free energy minima (labeled as I in Fig 1B) correspond largely to the well-defined dimers with full stacking overlap between the two interacting G-tetrads (ca. 2.2 nm², see S4 Fig) providing maximum stacking stabilization. More specifically, in the dominant 5-5 G-mediated state (5-5,I in Fig 2B), the relative polarity of the stacked G-tetrads allows for the largest possible stacking interaction between guanines with the pyrimidine (i.e., 6-member)

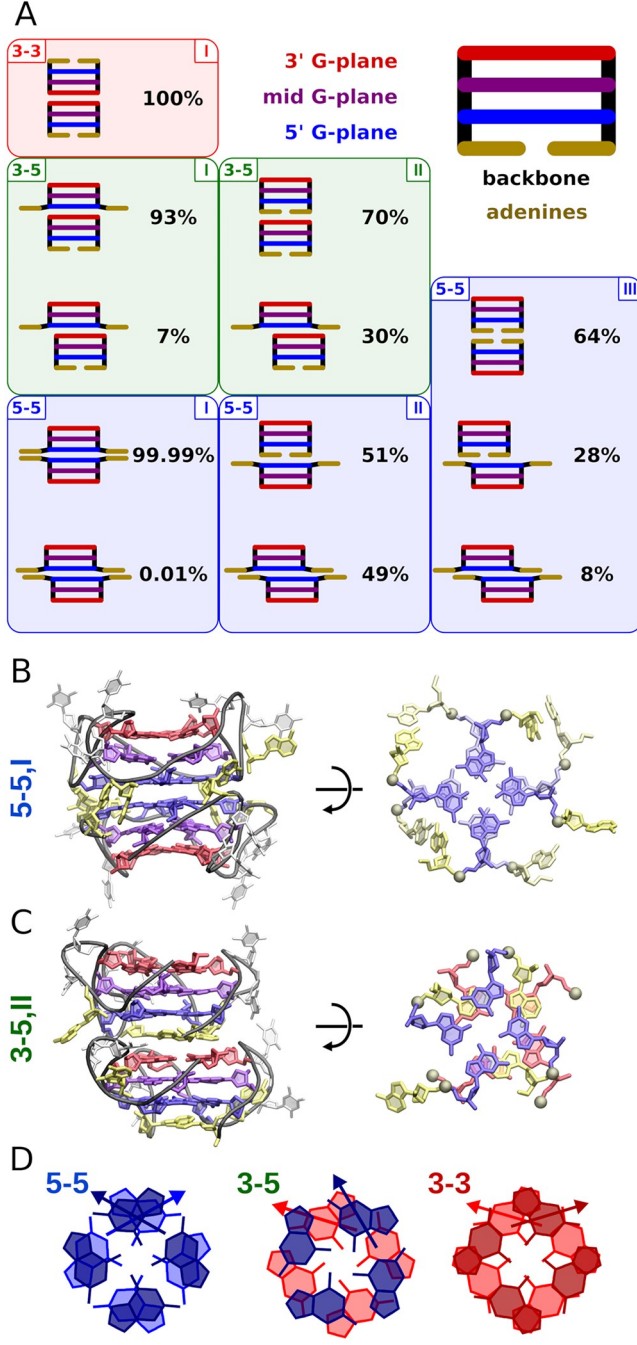

**Fig 2. Structural properties of G4 dimers.** A: Schematic representation of the G4 dimeric assemblies identified by the cluster analysis in all minima labeled in Fig 1B; corresponding equilibrium fractions, reported separately for each of the minima, were obtained from the properly unbiased REUS data. Adenine and thymine residues are shown in yellow and gray, respectively. B: Side and top view of the stacking interface in the G-mediated 5-5 mode (the 5-5,I minimum). C: Side and top view of the stacking interface in the A-mediated 3-5 mode (the 3-5,II minimum). Representative structures of the remaining states defined in Fig 1B are shown in S5 Fig. D: Average relative arrangement of the guanine residues and their dipole moments across the three stacking interfaces, determined from the spatial densities of interfacial guanines shown in S6 Fig.

rings of one G-tetrad overlapping with the imidazole (5-member) rings of the other and vice versa (Fig 2D and S6 Fig). Additionally, in this alignment, dipole moments of stacked guanines are oriented roughly perpendicular with respect to each other leading to the most favorable electrostatic interaction between G-tetrads among the three possible dimerization modes (Fig 2D). Notably, the same arrangement of guanines at the stacking interface was previously found experimentally for the 5-5 dimers of parallel G-quadruplexes formed both in aqueous solution [28] and in the crystal structure [31], which further supports our predictions (see S7 Fig a detailed comparison of the structures). For the other G-mediated dimers, the stacked guanines either almost do not overlap (3-5,I) or contact only via their imidazole rings (3-3,I), suggesting less favorable interaction (Fig 2D).

Cluster analysis also revealed that, in the dominant 5-5 G-mediated mode, the two G-quadruplexes are twisted with respect to each other such that their 4-fold symmetric sugar-phosphate backbones form a regular alternating pattern (Fig 2B). In particular, they are spatially shifted by one half-period, i.e., $\sim 45°$ (S8 Fig), which ensures that all the phosphate groups in one monomer are kept, on average, at the maximum possible distance from their nearest counterparts in the other monomer (S9 Fig). This specific relative arrangement of monomers in the G-mediated dimer supposedly allows for minimization of electrostatic repulsion between the negatively charged backbone chains. Consequently, it was also observed in the crystal structure of parallel telomeric G-quadruplexes where it possibly contributes to the stability of the crystal lattice (see S8 Fig for comparison of the relative arrangement o backbones) [31]. In the 3-5 and 3-3 G-mediated dimers, the relative arrangements of the sugar-phosphate backbones are different and markedly less regular (S8 Fig) causing the neighboring phosphate groups on the two strands to be, on average, closer to each other compared to 5-5 (S9 Fig). This is especially visible for the 3-3 mode, where ca. 3 pairs of phosphates are less than 0.6 nm apart. At the same time, we did not observe any significant dimerization-induced changes in the loop regions and sugar puckerings in individual G4 monomers that could account for the above difference (see S10 and S11 Figs).

Interestingly, the structure of G-mediated dimers in 5-5 mode also suggests additional stabilization by the outer ring formed by adenine residues stacking across the interface (right panel of Fig 2B). Indeed, analysis of the adenine contact area confirms formation of, on average, ca. 2 cross-strand adenine-adenine contacts (S4 Fig). This finding agrees well with the X-ray and NMR data showing 2 and 3 adenine-adenine stacking interactions, respectively, between the associated parallel G-quadruplexes [27, 31].

Among the G-mediated structures identified by the cluster analysis, only a very small percentage—0.01% and 7% in the 5-5 and 3-5 binding mode, respectively—show a partial stacking overlap between the two G-tetrads (ca. 1.68–1.84 nm$^2$) which are shifted by ca. 0.41–0.49 nm with respect to each other (see S12 Fig). In contrast, the remaining free energy minima (II and III in Fig 1B) are much more structurally diverse. They mainly correspond to structures mediated by one (II, see Fig 2C) or two (III) adenine layers with ca. 2–3 adenines per layer stacked between G-quadruplexes (A-mediated states). However, in both minima, a significant fraction (30–49%) of structures is found in which G-quadruplexes are only partially stacked onto each other showing the contact area of ca. 0.89–1.84 nm$^2$ (see S4 and S12 Figs).

## Balance between interphosphate repulsion and counterion binding determines the relative stability of possible G4 dimers

To identify main driving forces of dimerization process and the energetic determinants of the stability of G-quadruplex dimers, we have determined the enthalpy of dimerization, $\Delta H$, and decomposed it, for each of the three stacking orientations, into contributions due to

interactions between individual structural elements of the system. The contributions, shown as interaction matrices in Fig 3A, involve both electrostatic and van der Waals energies and were calculated by averaging the enthalpy differences between dimeric and dissociated states (defined consistently with Fig 1B) over the unbiased ensembles.

Large positive values of the overall $\Delta H$ (in the range of 185–720 kcal/mol) suggest that the formation of the considered G4 dimers is enthalpically highly unfavorable and thus that it is driven by solvent-mediated entropic forces. The interaction matrices in Fig 3A demonstrate that the dominant force opposing the formation of the G-mediated dimers is a strong electrostatic repulsion of the two sugar-phosphate backbones ($\Delta H_{BB1-BB2} \approx 5.4$–$5.9 \times 10^3$ kcal/mol). This repulsion is effectively screened by potassium counterions ($\Delta H_{K-BB1/BB2} + \Delta H_{K-K} \approx -5.2$–$6.3 \times 10^3$ kcal/mol) that, in the dimeric state, tend to localize in newly-created binding sites constituted by regions of strongly negative electrostatic potential between the G4 backbones (see also discussion below). High absolute values of these nearly canceling contributions to $\Delta H$ indicate that the stability of G4 dimers involves a fine balance between competing electrostatic forces. This finding also provides a molecular-level explanation for the effect of salt concentration on the extent of G4 dimerization [26] and is consistent with previously reported participation of the loop-bound cations in stabilizing the folded state of G4 monomers [51]. Notably, dimerization does not significantly affect the $K^+$-binding properties of the central channels that remain mostly occupied by counterions. Occasional binding of $K^+$ in the additional binding site between the stacked G-tetrads is however less important for the stability of the dimers (with maximum contribution $\Delta H_{K-G1/G2} \approx -190$ kcal/mol seen for 3-5).

Quite unexpectedly, a very similar balance of interactions and the critical role of counterions in stabilization is also found for the A-mediated states (lower triangles in Fig 3A and S13 Fig), which are characterized by a greater separation between the sugar-phosphate chains. A lower stability of the A-mediated dimers (see Table 1) arises from the fact that the repulsion between phosphates is not offset by attractive interactions to the same extent as in the G-mediated case. Specifically, as seen from S13 Fig, the A-mediated dimers lack additional stabilization provided by $\pi$-stacking interaction between the adjacent G-tetrads ($\Delta H_{G1-G2} \approx -34$–$40$ kcal/mol in the G-mediated case) and by the thymine-backbone and adenine-backbone interactions ($\Delta H_{T/A-BB} \approx -10$–$100$ kcal/mol in the G-mediated case).

Energy decomposition in Fig 3 also shows that, despite a common compensation pattern, the three possible dimerization modes differ markedly in terms of individual contributions to $\Delta H$. In particular, from Fig 3B it can be seen that, in the dominant 5-5 dimer, interphosphate repulsion is considerably (by 9%) weaker than in 3-3 and 3-5, and thus might underlie the preference for the 5-5 dimerization mode. In order to examine molecular basis of the differences in interphosphate repulsion, we calculated the electrostatic energy density of cross-strand pairwise interactions between the phosphates as a function of the distance between them (Fig 4A and S14 Fig). Low energy density at small P–P distances ($<1.4$ nm) in the 5-5 dimer indicates that the relatively weak electrostatic repulsion observed in this case can be attributed to the small number of very unfavorable close contacts between the phosphate groups at the dimer interface. This is also reflected in the 3D distribution maps in Fig 4B showing the largest spatial separation of the interfacial phosphates in the 5-5 dimer. In contrast, the 3-3 and 3-5 energy density profiles reveal that the number of close-contact repulsive interactions between the interfacial phosphates in the two remaining less stable dimerization modes is clearly greater, which is also depicted in the 3D distributions in Fig 4B. The electrostatic potentials surrounding G4 monomers, computed using the Poisson-Boltzmann approach [53], independently confirm that the 5-5 mode is characterized by weakest phosphate-phosphate repulsion among the three dimers considered (S15 Fig).

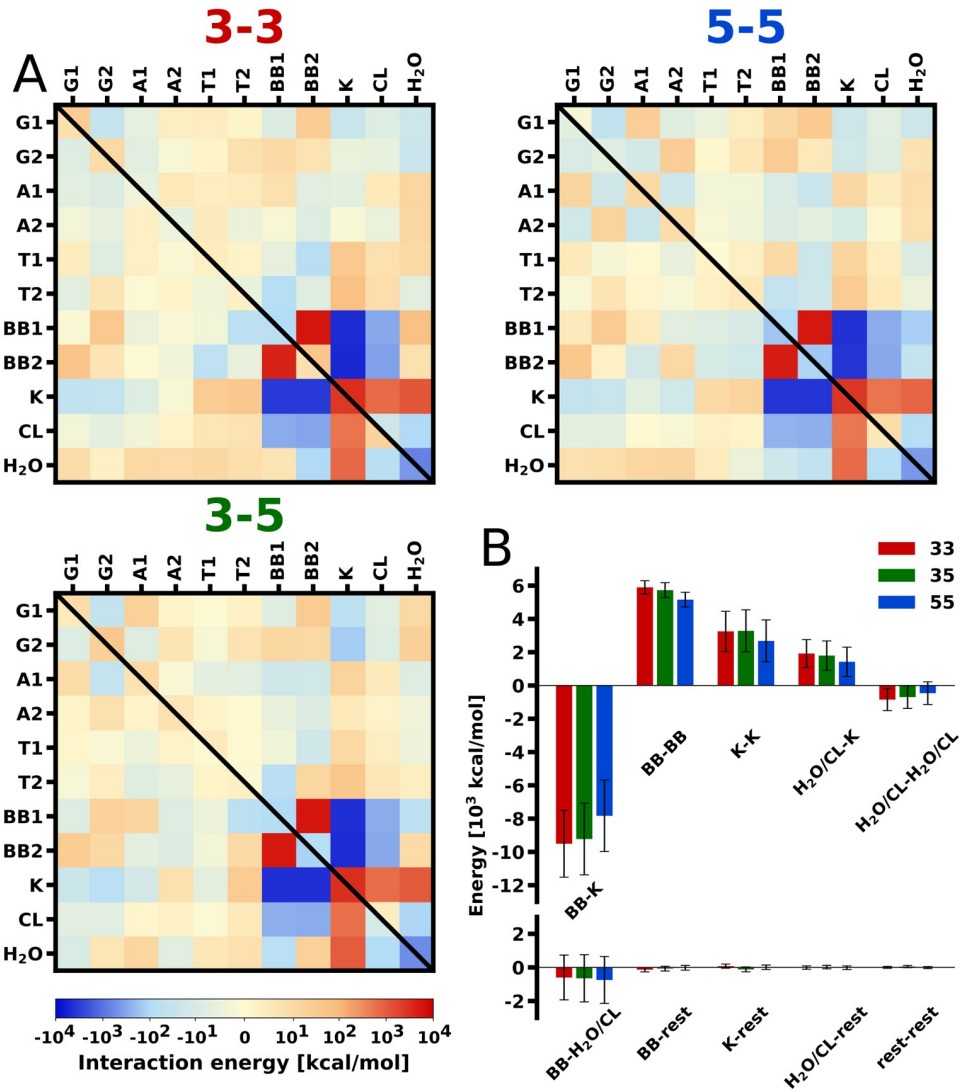

**Fig 3. Enthalpy of dimerization.** A: Detailed contributions to the dimerization enthalpy ($\Delta H$) computed as the interaction energy changes between individual structural elements of the system: guanines (G), adenines (A), thymines (T), backbone (BB), potassium cations (K), chloride anions (CL) and water ($H_2O$) in the G-mediated state (upper triangle matrix) and the A-mediated state (lower triangle matrix); individual G4 monomers are denoted as 1 and 2. B: Overall contributions to $\Delta H$ in the G-mediated dimers, without distinction between individual monomers.

Fig 3A indicates that some of the less dominant contributions to $\Delta H$ also favor the 5-5 dimerization mode. Notably, in this mode, the stacking interaction between the G-tetrads, $\Delta H_{G1\text{-}G2}$, is by $\sim 5$ kcal/mol more favorable than in both 3-3 and 3-5, which results from a more favorable alignment of guanine dipole moments at the 5-5 interface (see Fig 2D). Additional stabilization of the 5-5 dimer is provided by the cross-strand adenine-adenine interactions ($\Delta H_{A1\text{-}A2} \approx -18$ kcal/mol) that are missing in the 3-3 and 3-5 modes, as described above in the contact analysis (see Fig 2A and S4 Fig).

As shown by the inset in Fig 4A, the total interphosphate repulsion energy does not differ significantly between the 3-3 and 3-5 dimers, consistently with the values of $\Delta H_{BB1\text{-}BB2}$ in Fig 3A and $\Delta H_{BB\text{-}BB}$ in Fig 3B. This then raises a question as to why the 3-5 mode is found in our free energy simulations to be by $\sim 20$ kcal/mol more stable than 3-3. A hint comes from the

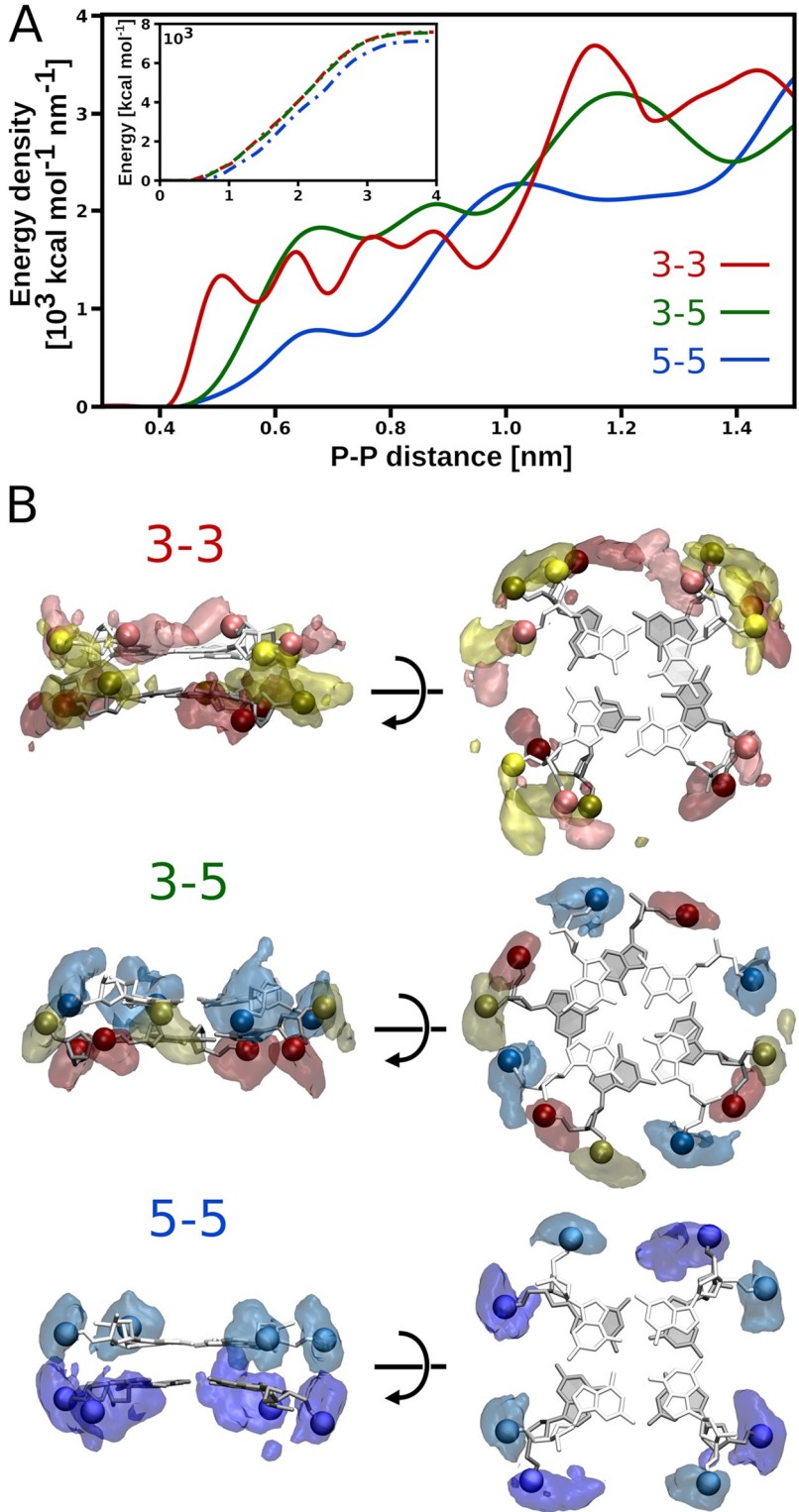

**Fig 4. Distribution and electrostatic repulsion of phosphates.** A: Electrostatic energy density of pairwise cross-strand interactions between the phosphate groups as a function of the distance between them (P–P distance; for its full range see S16 Fig). The inset shows cumulative interphosphate repulsion energy as a function of the P–P distance. B: Spatial distribution of all phosphate groups at the stacking interface that are located within 1 nm of any phosphate group of the other G4 monomer. The phosphate groups of the nucleotides in the 3'- and 5'-terminal G-tetrads are

shown in dark/light red and dark/light blue, respectively, while the remaining ones in yellow. For the corresponding parmbsc1 data see S14 Fig.

energy densities in Fig 4A, which show that association of parallel G4 units via their 3'-terminal G-tetrads leads to a number of energetically unfavorable van der Waals contacts between the phosphate groups (non-zero density at distances <0.55 nm). Since this destabilization of the 3-3 dimer cannot be sufficiently compensated for by counterion screening, this suggests that the difference in the stability between the 3-3 and 3-5 modes might arise from disparate ion binding properties.

## G4 dimerization equilibrium depends on the salt concentration and can be modulated by small-molecule ligands

The above analyses suggest that the differences in electrostatic screening of the phosphate groups at the stacking interface by $K^+$ counterions might be responsible for a significantly higher stability of the 3-5 dimer over the 3-3 dimer. To investigate these differences in more detail, we first re-computed the binding free energy profiles for all three dimers in the presence of $K^+$ counterions only, i.e., at the effective $K^+$ concentration lowered from the physiological concentration (0.15 M) to about 0.04 M.

From the free energy profiles in Fig 5A, it can be seen that, as the $K^+$ concentration is decreased nearly 4-fold, the dimers are destabilized by 2–7 kcal/mol, depending on the mode. This demonstrates directly that the presence of counterions is essential for G4 dimerization, in line with the experimental findings [26]. A 5 kcal/mol greater destabilization of the 3-5 than of the 3-3 mode confirms that $K^+$ binding at the dimeric interface does in fact underlie a much higher stability of the 3-5 dimer, as was expected from the phosphate-phosphate electrostatic energy density. Unexpectedly, however, a decrease in the number of $K^+$ ions at the 3-3 interface upon lowering the salt concentration (Fig 5B) is actually more pronounced than at the 3-5 interface (1.4 vs. 1.1, respectively, for $K^+$ present between the interfacial phosphates). This indicates that binding of individual $K^+$ ions at the 3-5 stacking interface has to be energetically more favorable, so as to provide greater stability to the 3-5 dimer.

The spatial distributions of $K^+$ ions at the three dimeric interfaces, shown in Fig 5C and S17 Fig, provide a possible explanation for the above differences in the $K^+$ binding strength. As can be seen, in the 3-5 mode, counterions are distributed non-uniformly at the stacking interface and penetrate deeper into the cleft between the two monomers, thereby being able to more effectively screen the electrostatic repulsion between the sugar-phosphate backbones. A large fraction of them (52%) is found in the three separate sites located between the pairs of opposite double-chain-reversal loops of the G4 units. These well-defined sites (Fig 5D), each formed by three phosphate groups—one from one G4 unit and two from the other—show a high cation-binding capacity and are on average occupied by ∼2.0 $K^+$ ions at a given time. In contrast, in the 3-3 mode, the $K^+$ distribution is more uniform, with interfacial ions spread over a larger area, which indicates less specific and weaker binding. Unlike in the two remaining dimers, in the 3-3 case, $K^+$ ions are mostly found outside the interface and penetrate less readily between the phosphates groups. The reason is that very closely separated phosphate groups at the 3-3 interface (Fig 4) do not allow counterions to bind between them and effectively mitigate the repulsive forces. The differences in hydration of the three interfaces are less pronounced; only in the 3-3 case, we observed a decreased hydration of the interfacial phosphates with respect to the monomeric state, probably due to above-mentioned involvement of some of the phosphates in direct interactions (S18 Fig).

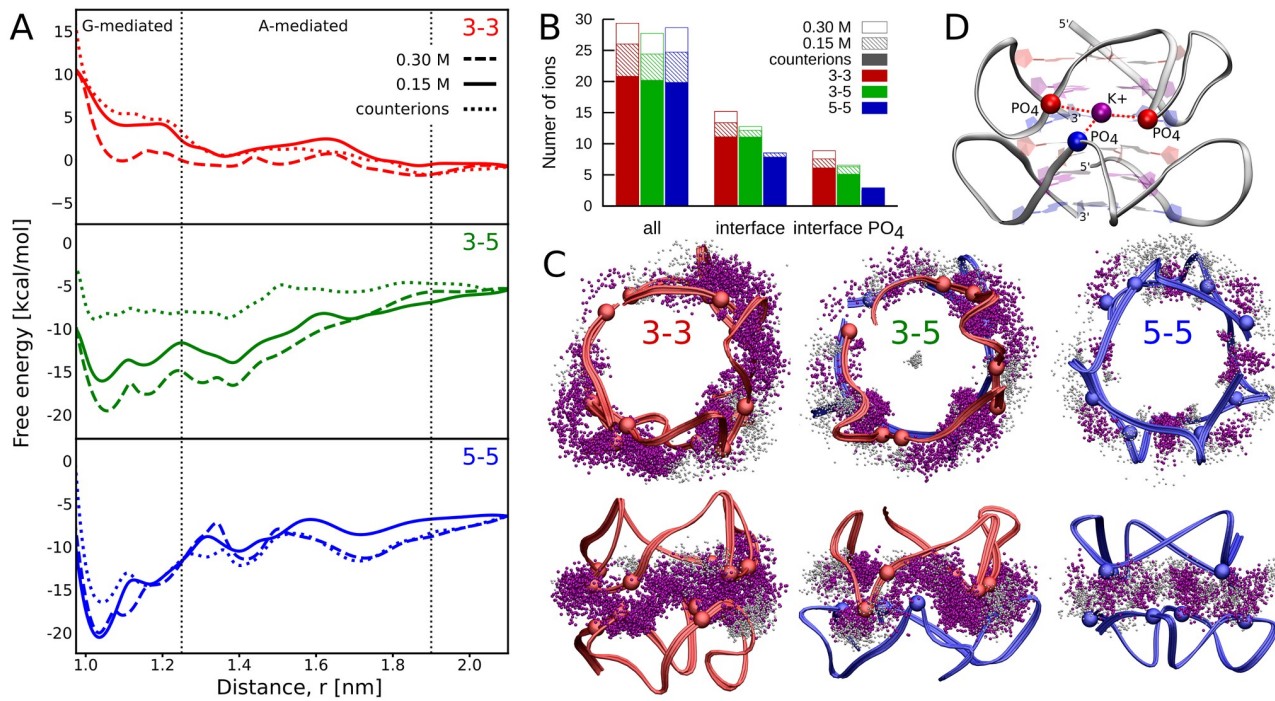

**Fig 5. Spatial distribution and influence of ions on G-quadruplexes dimerization.** A: Dependence of the free energy profiles for the formation of the 3-3, 3-5 and 5-5 dimers on the salt concentrations: only $K^+$ counterions (dotted lines), physiological KCl concentration of 0.15 M (solid lines) and 0.3 M KCl (dashed lines). B: Average numbers of $K^+$ ions in contact with the G-mediated dimers: within 0.7 nm of the backbones of any G4 unit (all), within 0.7 nm of both backbones at the same time (interface), within 0.7 nm of the $PO_4$ groups of the two G4 units at the same time (interface $PO_4$). C: Spatial distribution of $K^+$ ions at the stacking interface of each of the three dimerization modes, generated by superimposing 1,000 0.5-ns-separated snapshots at a physiological KCl concentration. The interface $K^+$ ions that at the same time are in contact with the $PO_4$ groups of both G4 units are highlighted in purple. For the corresponding parmbsc1 data see S17 Fig. D: Well-defined $K^+$-binding site formed by the three phosphates groups at the 3-5 interface.

The free energy profiles at increased concentration of counterions (0.3 M; Fig 5A) further confirm the observed differences in $K^+$-binding properties between the two dimers. Despite a 3–4 kcal/mol increase in the stability of the 3-5 mode, the number of $K^+$ ions at the interface does not increase concurrently (Fig 5B), indicating saturation of the binding sites already at lower concentrations. In turn, the 3-3 dimer remains unstable even at higher salt concentrations, even though the number of counterions non-specifically bound at the interface clearly increases.

Because the distances between the phosphate groups across the 5-5 interface are generally greater than 0.6 nm they do not form particularly favorable $K^+$-binding sites. As a result, the number of ions present at this interface is markedly lower than at the two remaining ones, with 28% of interfacial $K^+$ found in 8 low-affinity binding sites (Fig 5C). Notably, at the low salt concentration, the 5-5 dimer remains highly stable with $\Delta G^\circ_{bind} = -13.0$ kcal/mol (Fig 5A), which shows that the relatively small number of $K^+$ ions is sufficient for screening the interphosphate repulsion between the monomers in the 5-5 dimer. Accordingly, adding more counterions to the solution reduces the preference for the 5-5 dimer, which is most pronounced at the low concentration, when its binding sites are already almost saturated.

To test our conclusion that unfavorable interphosphate repulsion is responsible for destabilization of the two less preferred stacking modes, we recomputed the free energy profile for the formation of the most unfavorable 3-3 dimer, using a model aromatic ligand as a spacer to increase the distance between the interfacial phosphates. For this purpose, we used the

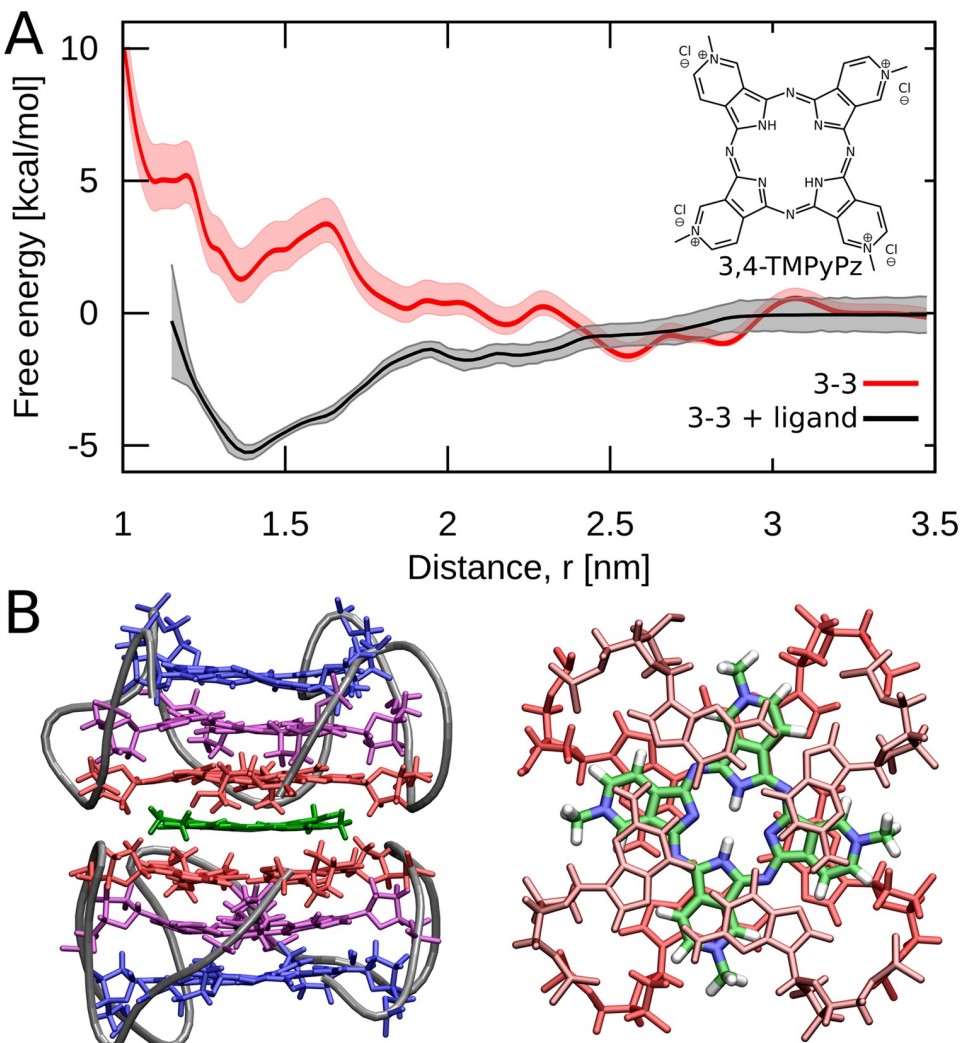

**Fig 6. Increased stability of the 3-3 dimer upon ligand binding.** A: Effect of the 3,4-TMPyPz ligand on the free energy profile for the formation of the 3-3 dimer. B: Representative structure of the 3-3 dimer mediated by 3,4-TMPyPz.

cationic 3,4-Tetramethylpyridiniumporphyrazine (3,4-TMPyPz, Fig 6A), which was shown to selectively bind to human telomeric G-quadruplex DNA by stacking on the G-tetrad surface [54, 55]. Because of the square shape, 3,4-TMPyPz mimics an additional aromatic plane between the two G4 units.

Fig 6A shows that 3,4-TMPyPz present between the 3'-terminal G-tetrads indeed ensures the stability of the dimer relative to the dissociated state. As assumed, this actually originates from increased separation distance between the interfacial phosphates that reduces the electrostatic energy density at small P–P distances compared to the ligand-free 3-3 mode or even the 3-5 mode, however, not to the degree observed in the locally-optimal 5-5 mode (S19 Fig). At the same time, the number of $K^+$ ions at the interface decreases by 66% relative to the ligand-free 3-3 case (see S20 Fig) which, given a non-optimal phosphate arrangement, may partially explain only a moderate increase in the dimer stability (by $\sim 6$ kcal/mol).

## Conclusions

In this work, we examined the molecular determinants of the dimerization of two parallel-stranded G-quadruplexes (G4) using molecular dynamics-based free energy calculations. We found that G-quadruplex dimers are formed almost exclusively (99.9%) by direct aggregation of the 5'-end G-tetrads (the G-mediated 5-5 mode), in agreement with all available experimental data [26–33]. At the same time, our simulations reveal that other dimeric G4 forms, particularly the 3-5 mode and even adenine-mediated dimers (A-mediated), are also thermodynamically stable relative to the dissociated state. An important exception is the G-mediated dimer formed by aggregation of the 3'-terminal G-tetrads (the 3-3 mode), which is found to be absent from the dimeric ensemble. The prevalence of the 5-5 dimer with the exposed 3'-terminal G-tetrads, that is unlikely to aggregate further, might explain a generally low tendency of G-quadruplexes to form higher-order oligomers [25, 56, 57].

Our results further indicate that, although the dimerization equilibrium is affected by a number of different enthalpic contributions, the observed aggregation propensities are primarily governed by a fine balance between the repulsion of the negatively-charged sugar-phosphate backbones and favorable counterion binding at the interface between the G4 monomers. Importantly, we found that the strong preference for the 5-5 mode results from a locally optimal arrangement of the backbones with largest inter-phosphate distances across the interface and, therefore, weakest electrostatic repulsion between the monomers. This finding provides an explanation for the experiment in which an additional phosphate group attached at the 5'-end of the parallel G-quadruplex prevents 5-5 dimerization [58]. In the two remaining less stable dimers, 3-5 and 3-3, electrostatic repulsion between the monomers is similar and significantly larger compared to the dominant 5-5 dimer. We found that a much higher stability of the 3-5 mode can be attributed to stronger counterion binding at the specific binding sites formed by the phosphate groups at the interface between the G4 units. Another notable contribution that promotes the 5-5 dimer turned out to be the stacking interaction between the 5'-terminal G-tetrads, which is characterized by almost maximum possible overlap and a favorable relative arrangement of the guanine dipole moments, as has been also suggested previously [45].

A molecular-level understanding of the G4 dimer formation process provided by our study suggests methods for shifting the dimerization equilibrium by controlling the electrostatic repulsion-attraction balance and thereby opens up new opportunities for designing oligomeric G-quadruplex structures. Specifically, we predict that by doubling the concentration of $K^+$ counterions we can significantly (from 0.1 to 16%) increase the equilibrium population of the 3-5 dimer which is especially suited to bind cations from aqueous solution. Moreover, our model indicates that the stability of the G-mediated dimers can be greatly enhanced by reducing the interphosphate repulsion which is the main destabilizing factor especially pronounced in the 3-3 and 3-5 cases. Indeed, increasing the separation distance between the interfacial phosphates by inserting a model aromatic ligand between the G4 units markedly improves the stability of the originally unstable 3-3 dimer relative to the monomeric state.

## Methods

### Simulation details

All simulated systems contained two monomers of the parallel G-quadruplexes with the human telomeric sequence d[AGGG(TTAGGG)$_3$], whose initial structure with two $K^+$ ions coordinated in the central channel was taken from the PDB database (PDB id 1KF1) [31].

G4 dimers were solvated with 22826 TIP3P water molecules [59] in dodecahedron box with a cell vector length of 10.1 nm, at physiological ionic strength (150 mM KCl) and, additionally, at two different salt concentrations, i.e., at 300 mM KCl, and at 40 mM which corresponds to 42 $K^+$ counterions only (including 2 ions in the central channel). The CHARMM36 force field [60] was used for DNA and ions because, as previously shown, it reproduces the structure and stability of parallel G-quadruplexes in aqueous solution [50, 61–64]. To test whether our conclusions are force-field independent, all dimerization simulations were additionally carried out using the AMBER parmbsc1 force field as a reference [65]. In additional simulations of the ligand-mediated dimer, the 3,4-TMPyPz ligand was parameterized using the CHARMM generalized force field (CGenFF) [66] and partial charges obtained from HF calculations in Gaussian [67] via Merz-Kollman ESP fitting, using the 6-31G* basis set (see S1 Table for numerical values).

The MD simulations were performed using Gromacs 5.0.4 [68] in the NPT ensemble, with the temperature kept at 300 K and using the v-rescale thermostat [69] and the pressure kept at 1 bar using Parrinello-Rahman barostat [70]. Periodic boundary conditions were applied in 3D, and electrostatic interactions were calculated using the particle mesh Ewald (PME) [71] method with a real-space cutoff of 1.2 nm and a Fourier grid spacing of 0.12 nm. A cut-off of 1.2 nm was used for Lennard-Jones interactions. All bond lengths were constrained using P-LINCS [72] for DNA and SETTLE [73] for water. The equations of motion were integrated using the leap-frog algorithm with a 2 fs time step.

## Free energy simulations

To study the relative stability of the dimers of the parallel G-quadruplexes with three different end-to-end stacking orientations (dimerization modes), we calculated the corresponding free energy profiles using replica exchange umbrella sampling (REUS). The calculations were carried out using the PLUMED 2.0 plugin [74] coupled to Gromacs. To generate the initial configurations for the REUS simulations, we first performed short equilibrium simulations (<250 ns) of the two G4 monomers initially fully separated, with the center of mass distance between their guanine cores (the reaction coordinate, $r$; see inset in Fig 1B) being in the range of 1.9–2.0 nm. Initially, the two monomers were oriented parallel to each other to promote their spontaneous association in one of the three considered dimerization modes. Indeed, in the case of the 3'-5' and 5'-5' modes, we observed spontaneous binding through base stacking interactions between the two interacting guanine planes (the G-mediated states) within 180 and 210 ns, respectively. To obtain the G-mediated state for the 3'-3' mode that did not form spontaneously, we ran an additional steered MD simulation, in which the G4 monomers were driven towards each other during 100 ns using a moving harmonic potential with a force constant of 286.1 kcal/(mol·nm$^2$) applied to the coordinate $r$. Next, to initially sample the full range of the reaction coordinate from the G-mediated bound state (1.0 nm) up to a fully unbound state (3.5 nm), for each of the dimers, the two G4 units were forced to dissociate over 100 ns by applying the same moving harmonic potential as above.

To ensure better convergence of the free energy profiles, in the critical range of $r$ from 1.0 to 2.0 nm, the initial frames were taken alternately from the association and enforced dissociation trajectories, whereas in the range of 2.0–3.5 nm from the latter only. We used 25 uniformly distributed 0.1-nm separated REUS windows. In each of these windows the system was simulated for 0.5 $\mu$s, using the harmonic potential with a force constant of 286.1 kcal/(mol·nm$^2$) to restrain the system along the reaction coordinate $r$. The exchanges between neighboring windows were attempted every 2 ps and the acceptance rate turned out to be ~21%. No additional restraints were applied on the relative orientation of the associating G4

units. The free energy profiles were determined from the last 450 ns of thus obtained trajectories using the standard weighted histogram analysis method (WHAM 2.0.9) [75]. Uncertainties were estimated using bootstrap error analysis taking into account the correlation in the analyzed ftime series. The dimerization free energies were estimated from the free energy profiles using a simple approach:

$$\Delta G^{\circ}_{dim} = -k_B T \ln \left( \frac{1}{V_0} \int_0^R dr \int_0^\pi d\theta \int_0^{2\pi} d\phi \, r^2 \sin\theta \exp\left(-\frac{G(r)}{k_B T}\right) \frac{1}{4\pi r^2} \right) \tag{1}$$

where $G(r)$ is the free energy profile taken directly from the REUS calculations, $(4\pi r^2)^{-1}$ is a radial correction term, $V_0$ is the standard volume (1661 Å$^3$) corresponding to the standard concentration of 1 M, $R$ is the upper distance limit defining the dimeric state, $k_B$ is the Boltzmann constant and $T$ the temperature [76, 77]. All molecular images were created using VMD 1.9.2 [78].

## Supporting information

**S1 Fig. Distributions of the relative orientations between G-quadruplexes in the three considered dimerization modes as a function of the center of mass distance between the two G4 units.** The orientation angle between G4 units is defined as the angle between two vectors passing through the center of G-tetrads and perpendicular to them. It can be seen that G-quadruplexes dimerize through cofacial stacking (parallel orientation between the interfacial G-tetrads at short distances, with the orientation angle $\approx 180°$ or $0°$ for 5-5 and 3-3 or 3-5 dimers, respetively). At longer distances the systems sample the entire range of relative orientations between G-quadruplexes (the orientation angle in the range of 0–180°).
(TIF)

**S2 Fig. Convergence of the free energy profiles for the formation of the three considered dimers with the simulation time.**
(TIF)

**S3 Fig. Dimerization preferences obtained with the AMBER parmbsc1 force field.** Free energy profiles for the formation of the dimers computed using parmbsc1 as a function of the separation distance between them, $r$. Even though dimer formation is consistently less favorable than in the CHARMM36 simulations, the stacking preferences are the same.
(TIF)

**S4 Fig. Equilibrium distributions of the contact area between guanines and adenines at the stacking interfaces in all the states defined in Fig 1B.** Different colors are used to map the distributions on the structural clusters produced by our cluster analysis.
(TIF)

**S5 Fig. Side and top view of the stacking interfaces for all dimeric states defined within the 3-3, 3-5 and 5-5 modes.**
(TIF)

**S6 Fig. Spatial densities of the guanine residues at the stacking interface of the 3-3, 3-5 and 5-5 modes.**
(TIF)

**S7 Fig. Comparison with x-ray data.** Structural comparison of the 5-5 dimer predicted by our MD simulations (red, 5-5,I in Fig 1B) with the 5-5 stacking arrangement of parallel telomeric G-quadruplexes found in the x-ray structure (blue, PDB: 1KF1). The structures were

superimposed based on the positions of the phosphate groups. The average heavy-atom RMSD between the MD snapshots and the x-ray structure is 0.55 nm.
(TIF)

**S8 Fig. Relative arrangement of the sugar-phosphate backbones of the associated G4 units in the three G-mediated dimerization modes.** Relative arrangement presented as projections of the atomic positions of backbones on the curved surface of the cylinder with a diameter corresponding to the G-quadruplex dimer.
(TIF)

**S9 Fig. Distribution of shortest interphosphate distances (minimal P–P distance) across the dimeric interface in the three considered dimerization modes.**
(TIF)

**S10 Fig. Comparison of the percentage populations of sugar puckers between the monomeric G4 and the three dimer interfaces.**
(TIF)

**S11 Fig. Spatial range of structural fluctuations of the sugar-phosphate backbone of the monomeric and dimeric G4.** Isosurfaces show 90% of the total fluctuation range. For each G4 monomer one side and two top views (from the 5'- and 3'-ends) are shown.
(TIF)

**S12 Fig. Equilibrium distributions of the shift coordinate.** Shift is defined as the distance between the axes of the G4 units projected on the stacking interface plane. The axes were determined as vectors connecting the centers of mass of the two external G-tetrads of each of the G4 units. Different colors are used to map the distributions on the structural clusters produced by our cluster analysis.
(TIF)

**S13 Fig. Enthalpy of dimerization.** Detailed contributions to the dimerization enthalpy ($\Delta H$) computed as the interaction energy changes between individual structural elements of the system: guanines (G), adenines (A), thymines (T), backbone (BB), potassium cations (K), chloride anions (CL) and water ($H_2O$) in the G-mediated state (upper triangle matrix) and the A-mediated state (lower triangle matrix); individual G4 monomers are denoted as 1 and 2. For clarity, numeric values were rounded to the nearest integer.
(TIF)

**S14 Fig. Electrostatic interactions and spatial distribution of the sugar-phosphate backbones.** A) Electrostatic energy density of pairwise cross-strand interactions between the phosphate groups as a function of the distance between them in the parmbsc1 force field simulations. The inset shows cumulative interphosphate repulsion energy as a function of the P–P distance. B) Spatial distribution of all phosphate groups at the stacking interface that are located within 1 nm of any phosphate group of the other G4 monomer in the parmbsc1 force field simulations. The phosphate groups of the nucleotides in the 3'- and 5'-terminal G-tetrads are shown in red and blue, respectively, while the remaining ones in yellow.
(TIF)

**S15 Fig. Distribution of the electrostatic potential ($-10$ kT/e) generated by a single G4 unit in the three studied dimers.** Distribution obtained by integrating the linearized Poisson Boltzmann equation using Adaptive Poisson-Boltzmann Solver (APBS) (N. A. Baker, et al. Proc. Natl. Acad. Sci. U.S.A. 98, 10037, 2001). The concentrations of $+1$ and $-1$ ion species were set to 150 mM with an ion exclusion radius of 0.2 nm. Grid size of $120^3$ with grid spacing of 0.069

nm was used. We used the Dirichlet boundary conditions with the boundary potential value determined from a Debye-Huckel model for a single sphere with a point charge, dipole, and quadruple. Blue balls in the top panel represent the positions of the phosphate groups of the opposite G4 unit. Blue dots in the bottom panel show the distribution of all heavy atoms in the adjacent guanine plane of the opposite G4 unit.
(TIF)

**S16 Fig. Electrostatic energy density of pairwise cross-strand interactions between the phosphates as a function of the distance between them (full range of the P–P distance).**
(TIF)

**S17 Fig. Spatial distribution of $K^+$ ions at the stacking interface of each of the three dimerization modes.** Generated by superimposing 1,000 0.5-ns-separated snapshots at a physiological KCl concentration from the parmbsc1 force field simulations. The interface $K^+$ ions that at the same time are in contact with the $PO_4$ groups of both G4 units are highlighted in purple.
(TIF)

**S18 Fig. Spatial distribution of water molecules at the stacking interface of each of the three dimerization modes.** Generated by superimposing 100 5-ns-separated snapshots at a physiological KCl concentration. The plot shows radial distribution functions of water molecules with respect to the phosphate groups of the nucleotides in the external G-tetrads (monomer) or the G-tetrads involved in dimer formation (dimers).
(TIF)

**S19 Fig. Phosphate-phosphate repulsion is reduced upon ligand binding.** Effect of the 3,4-TMPyPz ligand intercalated between the two G4 units on the electrostatic energy density of pairwise cross-strand interactions between the phosphates as a function of the distance between them (P–P distance). The inset shows cumulative interphosphate repulsion energy as a function of the P–P distance.
(TIF)

**S20 Fig. Effect of the 3,4-TMPyPz ligand on the spatial distribution of $K^+$ ions at the stacking interface of the 3-3 dimer.** A) The images were generated by superimposing 1,000 0.5-ns-separated snapshots at a physiological KCl concentration. The interface $K^+$ ions that at the same time are in contact with the $PO_4$ groups of both G4 units are highlighted in purple. B) Average numbers of $K^+$ ions in contact with the G-mediated dimers: within 0.7 nm of the backbones of any G4 unit (all), within 0.7 nm of both backbones at the same time (interface), within 0.7 nm of the $PO_4$ groups of the two G4 units at the same time (interface $PO_4$).
(TIF)

**S1 Table. Partial charges obtained for 3,4-TMPyPz ligand.**
(PDF)

## Author Contributions

**Conceptualization:** Mateusz Kogut, Jacek Czub.

**Formal analysis:** Mateusz Kogut, Cyprian Kleist.

**Supervision:** Jacek Czub.

**Validation:** Jacek Czub.

**Visualization:** Mateusz Kogut, Cyprian Kleist.

**Writing – original draft:** Mateusz Kogut, Jacek Czub.

**Writing – review & editing:** Mateusz Kogut, Jacek Czub.

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
