## [Decision Letter · Decision Letter 0]

18 Jul 2019

Dear Dr Czub,

Thank you very much for submitting your manuscript 'Why do G-quadruplexes dimerize through the 5’-ends? Driving forces for G4 DNA dimerization examined in atomic detail.' for review by PLOS Computational Biology. Your manuscript has been fully evaluated by the PLOS Computational Biology editorial team and in this case also by independent peer reviewers. The reviewers appreciated the attention to an important problem, but raised some substantial concerns about the manuscript as it currently stands. While your manuscript cannot be accepted in its present form, we are willing to consider a revised version in which the issues raised by the reviewers have been adequately addressed. We cannot, of course, promise publication at that time.

Sincerely,

Guanghong Wei

Associate Editor

PLOS Computational Biology

Nir Ben-Tal

Deputy Editor

PLOS Computational Biology

[LINK]

Reviewer's Responses to Questions

**Comments to the Authors:**

Reviewer #1: In this work the authors studied the stacking patterns of two G-quadruplexes using large-scale replica exchange MD simulations. Based on the simulations, they gave the dominant stacking model, analyzed the corresponding structures in detail, and investigated the underlying interactions for the stability. They also studied the binding free energies as a function of salt concentrations. The knowledge gained in the study may facilitate controlling G4 assembling and designing new G4 oligomers. Overall, I found the simulations were well done, the analysis was sufficient, and the results were solid. I suggest with acceptance of the paper for publication, provided the following minor problems are fixed.

1) Figure captions are not complete, e.g., what are the shaded regions in Fig. 1, what are the color codes in Fig. 2 and 4?

2) In page 9, line 375, the author mentioned “with 42 K+ counterions …”The sentence is hard to understand. And what the equivalent concentration of this “with 42 K+ conterions”?

3) There are some relevant works published previously, e.g., Y. Bian, et al., PLOS CB 2014, v10, e1003562, and J. Zhou, et al., JACS 2017, v139, 7768, the author may want to discuss their relevance with this work.

Reviewer #2: The paper by Czub et al. is a computational study of preferential orientations of GQ dimerization process. From the well-converged free energy calculation as a function of a distance between two guanine cores, they found that G-quadruplexes preferentially dimerize through the 5'-ends. The reason for this 5'-end preference is due to fine balance between interphosphate repulsion and counterion binding. This paper is well written and demonstrates a power of molecular dynamics simulation methods. I would recommend this paper for the publication of PLOS Comp. Biol. after minor revisions.

1. Although the authors pointed out an important role of the enthalpic contribution for such dimerization process and decomposed the enthalpic contributions into several factors, it is difficult to find the total enthalpic and entropic changes in the manuscript. These total enthalpic and entropic changes need to be specified at the first place to clearly state whether the dimerization is due to enthalpic or entropic origin.

2. The authors concluded that 5'-end binding is preferred mainly due to less interphosphate repulsion. Are there any indications of sugar puckering changes for 5'-end, 5’-3’ end, 3’-end dimerization cases to modulate such interphosphate repulsive interactions?

3. In the dimerization via 5'-end, 5’-3’ end, or 3’-end, are there any distinctive perturbation of loop regions?

4. In the GQ dimer, spatial distribution of potassium ions was plotted. In comparison with ion distribution, it would be nice to plot spatial distribution of water at the binding interface to gauge any possible dehydration effect after the dimerization.

5. For 3,4-TMPyPz ligand, which was used for the 3-3 dimer, the partial charges they derived need to be included in SI.

Reviewer #3: Please see the attached review.

**Have all data underlying the figures and results presented in the manuscript been provided?**

Reviewer #1: Yes

Reviewer #2: Yes

Reviewer #3: Yes

PLOS authors have the option to publish the peer review history of their article (what does this mean?). If published, this will include your full peer review and any attached files.

Reviewer #1: No

Reviewer #2: No

Reviewer #3: Yes: Nanjie Deng

---

## [Decision Letter · Decision Letter 1]

29 Aug 2019

[EXSCINDED]

Dear Dr Czub,

Thank you very much for submitting your manuscript, 'Why do G-quadruplexes dimerize through the 5’-ends? Driving forces for G4 DNA dimerization examined in atomic detail.', to PLOS Computational Biology. As with all papers submitted to the journal, yours was fully evaluated by the PLOS Computational Biology editorial team, and in this case, by independent peer reviewers. The reviewers appreciated the attention to an important topic but identified some aspects of the manuscript that should be improved.

We would therefore like to ask you to modify the manuscript according to the review recommendations before we can consider your manuscript for acceptance. Your revisions should address the specific points made by each reviewer and we encourage you to respond to particular issues Please note while forming your response, if your article is accepted, you may have the opportunity to make the peer review history publicly available. The record will include editor decision letters (with reviews) and your responses to reviewer comments. If eligible, we will contact you to opt in or out.raised.

- Supporting Information uploaded as separate files, titled 'Dataset', 'Figure', 'Table', 'Text', 'Protocol', 'Audio', or 'Video'.

We hope to receive your revised manuscript within the next 30 days. If you anticipate any delay in its return, we ask that you let us know the expected resubmission date by email at ploscompbiol@plos.org.

Sincerely,

Guanghong Wei

Associate Editor

PLOS Computational Biology

Nir Ben-Tal

Deputy Editor

PLOS Computational Biology

[LINK]

Reviewer's Responses to Questions

**Comments to the Authors:**

Reviewer #1: The authors have answered all my questions, I suggest the acceptance of the manuscript for publication.

Reviewer #2: none

Reviewer #3: Please see the attached review.

**Have all data underlying the figures and results presented in the manuscript been provided?**

Reviewer #1: Yes

Reviewer #2: Yes

Reviewer #3: Yes

PLOS authors have the option to publish the peer review history of their article (what does this mean?). If published, this will include your full peer review and any attached files.

Reviewer #1: No

Reviewer #2: No

Reviewer #3: Yes: Nanjie Deng

---

## [Decision Letter · Decision Letter 2]

9 Sep 2019

Dear Dr Czub,

We are pleased to inform you that your manuscript 'Why do G-quadruplexes dimerize through the 5’-ends? Driving forces for G4 DNA dimerization examined in atomic detail.' has been provisionally accepted for publication in PLOS Computational Biology.

In the meantime, please log into Editorial Manager at https://www.editorialmanager.com/pcompbiol/, click the "Update My Information" link at the top of the page, and update your user information to ensure an efficient production and billing process.

One of the goals of PLOS is to make science accessible to educators and the public. PLOS staff issue occasional press releases and make early versions of PLOS Computational Biology articles available to science writers and journalists. PLOS staff also collaborate with Communication and Public Information Offices and would be happy to work with the relevant people at your institution or funding agency. If your institution or funding agency is interested in promoting your findings, please ask them to coordinate their releases with PLOS (contact ploscompbiol@plos.org).

Thank you again for supporting Open Access publishing. We look forward to publishing your paper in PLOS Computational Biology.

Sincerely,

Guanghong Wei

Associate Editor

PLOS Computational Biology

Nir Ben-Tal

Deputy Editor

PLOS Computational Biology

Reviewer's Responses to Questions

**Comments to the Authors:**

Reviewer #3: The authors have addressed my concern and I think it should be published without further delay. One more correction, Page 3, line 99, "Table ??" should be "Table 1".

**Have all data underlying the figures and results presented in the manuscript been provided?**

Reviewer #3: Yes

PLOS authors have the option to publish the peer review history of their article (what does this mean?). If published, this will include your full peer review and any attached files.

Reviewer #3: No

---

## [Editor Report · Acceptance letter]

13 Sep 2019

PCOMPBIOL-D-19-00976R2 

Why do G-quadruplexes dimerize through the 5’-ends? Driving forces for G4 DNA dimerization examined in atomic detail

Dear Dr Czub,

I am pleased to inform you that your manuscript has been formally accepted for publication in PLOS Computational Biology. Your manuscript is now with our production department and you will be notified of the publication date in due course.

With kind regards,

Matt Lyles
